# The Effect of Psychological First Aid Training on Knowledge and Understanding about Psychosocial Support Principles: A Cluster-Randomized Controlled Trial

**DOI:** 10.3390/ijerph17020484

**Published:** 2020-01-11

**Authors:** Marit Sijbrandij, Rebecca Horn, Rebecca Esliker, Fiona O’May, Relinde Reiffers, Leontien Ruttenberg, Kimberly Stam, Joop de Jong, Alastair Ager

**Affiliations:** 1Department of Clinical, Neuro and Developmental Psychology, World Health Organization Collaborating Center for Research and Dissemination of Psychological Interventions, Amsterdam Public Health Research Institute, Vrije Universiteit, 1081 BT Amsterdam, The Netherlands; 2Institute for Global Health and Development, Queen Margaret University, Edinburgh EH21 6UU, UK; rhorn@qmu.ac.uk (R.H.); fionaomay@gmail.com (F.O.); AAger@qmu.ac.uk (A.A.); 3Department of Counselling Psychology, University of Makeni, Freetown, Makeni PO Box 2, Northern Province, Sierra Leone; ebesliker@gmail.com; 4ARQ International, ARQ National Psychotrauma Centre, 1112 XE Diemen, The Netherlands; R.Reiffers@wartrauma.nl (R.R.); kimberlystam@gmail.com (K.S.); 5International Medical Relief Services (IMRES), 8200 AE Lelystad, The Netherlands; Ruttenberg@imres.nl; 6Amsterdam University Medical Centre, 1081 HV Amsterdam, The Netherlands; jtvmdejong@gmail.com; 7Department of Population and Family Health, Mailman School of Public Health, Columbia University, New York, NY 10032, USA

**Keywords:** psychological first aid, mental health and psychosocial support, humanitarian assistance, randomized controlled trial, prevention, training classification

## Abstract

Psychological first aid (PFA) is a world-wide implemented approach to helping people affected by an emergency, disaster, or other adverse event. Controlled evaluations of PFA’s training effects are lacking. We evaluated the effectiveness of a one-day PFA training on the acquisition and retention of knowledge of appropriate responses and skills in the acute aftermath of adversity in Peripheral Health Units (PHUs) in post-Ebola Sierra Leone. Secondary outcomes were professional quality of life, confidence in supporting a distressed person, and professional attitude. PHUs in Sierra Leone (*n* = 129) were cluster-randomized across PFA (206 participants) and control (202 participants) in March 2017. Knowledge and understanding of psychosocial support principles and skills were measured with a questionnaire and two patient scenarios to which participants described helpful responses. Professional attitude, confidence, and professional quality of life were assessed using self-report instruments. Assessments took place at baseline and at three- and six-months post-baseline. The PFA group had a stronger increase in PFA knowledge and understanding at the post-PFA training assessment (d = 0.50; *p* < 0.001) and at follow-up (d = 0.43; *p* = 0.001). In addition, the PFA group showed better responses to the scenarios at six-months follow-up (d = 0.38; *p* = 0.0002) but not at the post-assessment (d = 0.04; *p* = 0.26). No overall significant differences were found for professional attitude, confidence, and professional quality of life. In conclusion, PFA training improved acquisition and retention of knowledge and understanding of appropriate psychosocial responses and skills in providing support to individuals exposed to acute adversity. Our data support the use of PFA trainings to strengthen capacity for psychosocial support in contexts of disaster and humanitarian crisis. Future studies should examine the effects of PFA on psychosocial outcomes for people affected by crises.

## 1. Introduction

Psychological first aid (PFA) is an approach for assisting people in the immediate aftermath of disaster and humanitarian crisis to reduce initial distress and to foster short and long-term adaptive functioning [1]. PFA is largely intended for use by helpers in contact with people recently impacted by distressing events. They may include staff or volunteers such as health workers not specialized in mental health and psychosocial support, community health workers, and teachers [2].

PFA consists of assessing needs and concerns, helping people to address basic needs, listening without pressuring people, comforting people, linking people to information, services, and social supports, and protecting them from further harm [3]. PFA was originally designed as an evidence-informed approach to replace the widely implemented debriefing method, which has been shown to increase the risk for development of adverse psychological outcomes, such as symptoms of posttraumatic stress disorder (PTSD) [4,5]. Contrary to debriefing, PFA does not necessarily involve a discussion of the event that caused the distress [6,7]. Instead, it is designed to promote elements that, according to expert consensus [8], are most crucial in the early aftermath of crisis. These elements are a sense of safety, calm, self-and community efficacy, connectedness, and hope [8].

PFA trainings are delivered worldwide across high income countries (e.g., disaster relief organizations, military, etc.) [9] and in humanitarian and disaster contexts in low and middle income countries [10]. PFA training or orientation takes place over a period between one-half or, more typically, a full day, and includes explanations of the basic concepts and PFA principles, including how to approach a situation safely, how to support very distressed people, and how not to cause further harm [2]. It uses participatory learning including role-play. PFA trainings are considered to enhance knowledge and competence with regard to several domains, including establishing initial contact, helping responsibly, preparation for helping, observing the situation, listening, referral, and self-care [11].

Despite worldwide consensus that PFA should be the first-line approach in the immediate aftermath of disaster and humanitarian crisis [7,12], controlled evaluations of the effects of PFA training in early responders are still unavailable. A small pilot study of 82 school counsellors in Korea given a one-day PFA training suggested positive pre- to post-effects on PFA knowledge, perceived competence in PFA skills, and perceived preparedness and confidence to provide psychological assistance for future disasters [13]. However, this study did not include a control group of counsellors not receiving PFA, raising uncertainty about the added effects of PFA training on knowledge of psychological support strategies. Another uncontrolled study indicated that participation in PFA training was perceived to increase confidence in working with adults and children in the aftermath of two hurricanes in the United States [9].

Between 2014 to 2016, West African countries were struck by a major outbreak of Ebola virus disease (EVD) [14]. Widespread PFA capacity building of Ebola response teams (health personnel, inter-sectoral responders such as contact tracers, community members, or teachers) occurred during all phases of the Ebola outbreak. PFA capacity building is still ongoing in West Africa in an effort to further strengthen public health systems and disaster preparedness.

The primary aim of this study was to extend the evidence base concerning PFA by evaluating the effectiveness of a one-day PFA training on the acquisition and retention of knowledge of appropriate psychosocial responses and skills in the acute aftermath of adversity in Peripheral Health Units (PHU) staff members across Sierra Leone. Secondarily, we investigated whether PFA training improved professional quality of life, confidence in supporting a distressed person, and professional attitude.

## 2. Materials and Methods

### 2.1. Participants and Procedures

The study was carried out by Queen Margaret University, United Kingdom, in collaboration with the War Trauma Foundation and VU University Amsterdam, The Netherlands, University of Makeni, Sierra Leone and John Snow Research & Training Institute, Inc. (the latter not involved in analysis or interpretation of the data). This cluster-randomized controlled trial (cRCT) in Sierra Leone was part of a larger mixed-methods study on the effectiveness of the PFA training conducted in Liberia and Sierra Leone during this period [15]. Ethics approval was obtained from Queen Margaret University and the Government of Sierra Leone (03 August 2016).

At the time the study was conceived, no data on the expected effects of PFA trainings on retention of PFA principles were available. In order to identify a small to medium effect of 0.20 on retention of PFA principles, we calculated a required sample size of 260 (power = 0.95, *p* = 0.05, two-sided). Taking into account 20% attrition at follow-up, we aimed to include 312 individuals. Within Sierra Leone, PHUs consist of an average number of two to three primary health care workers. Therefore, we aimed to include at least 126 PHUs (63 randomized to the PFA program group; 63 randomized to control). We approached staff members for participation across a total number of 143 PHUs across 6 districts of Sierra Leone (Bombali, Kailahun, Kenema, Port Loko, Western Rural, Western Urban).

Primary health care workers (age ≥18 years) were included in the study if they: (1) had adequate oral and written command of the English or Krio language; and (2) had not previously received any PFA training or a training with overlapping content (i.e., they were PFA naïve). All participants were asked for oral and written informed consent to participate in the study.

Demographic information was gathered at the start of the assessments. Next, self-report questionnaires were administered to assess knowledge of appropriate responses to highly distressed individuals on how to apply psychosocial skills, professional attitude, confidence, and professional quality of life. The baseline assessment took place in the period between March 6 to April 6 in 2017, a 3-month post-assessment between June 28 and July 18 (on average 14.1 (SD = 1.2)) weeks after the baseline, and a 6-month follow-up assessment between October 2 and December 10, 2017 (at on average 27.8 (SD = 1.1)) weeks after the baseline. The 3-month post-assessment was timed to follow shortly after the receipt of the PFA training by the PFA group; the 6-month follow-up served as a measure of retention and of consolidation post-training. Measures were administered in the English language since the health workers participating all had adequate command of English and were trained in English. Twelve trained assessors, who were blind to the group participants were assigned to, administered the instruments.

### 2.2. Measures

Demographic characteristics assessed comprised age, gender, district, religion, educational level, years of job experience, and whether or not the participants had direct contact with acutely distressed patients or their relatives as part of their daily job.

Knowledge about psychosocial support for individuals who are exposed to adversities in line with the PFA principles was assessed with a study-specific overall knowledge test comprising 16 multiple choice questions with four response options, 7 items with two response options (yes/no) and 2 open-ended questions. None of these questions referred explicitly to PFA to preclude an advantage of PFA participants over the control participants. An example of a multiple choice question was: “What would be helpful to say to someone whose house burnt down? (a) I’m so sorry this has happened to you; (b) Did you cause the fire?; (c) You’re lucky, you could have died!; (d) Don’t worry, I know everything will be fine.” For the multiple choice questions, only one answer was judged correct. The four open-ended questions asked about signs of good listening, what actions should be taken if a person tries to hurt themselves or someone else, stress management strategies used by the respondent, and different ways in which a person can react after experiencing a distressing event. The items were scored by pairs of independent raters and recalculated into weighted scores on a 1–10 scale. Interrater reliability (intraclass correlation coefficient; ICC) for the four questions was adequate, varying between 0.69 to 0.96. To assess overall retention of PFA knowledge, we added the total score of the 16 multiple choice items, the 7 yes/no questions, and the mean of the weighted scores of the 4 open-ended questions (score range overall retention scale: 1 to 26). The total scale showed unsatisfactory reliability (*α* = 0.21 at baseline, *α* = 0.20 at post-assessment, and *α* = 0.20 at follow-up). After omitting item 16 (“How can you understand if someone is very seriously distressed?”), Cronbach’s alpha improved to *α* = 0.52 at baseline, *α* = 0.57 at post-assessment, and *α* = 0.59 at follow-up. 

Understanding of how to apply appropriate skills and response strategies for individuals who are exposed to adversities was assessed by presenting two case scenarios of patients presenting in the early aftermath of a very distressing event. Participants were asked to describe their professional response to each scenario with open-ended questions, and their responses were recorded. Responses were scored on a 0–10 (scenario 1) and 0–4 (scenario 2) scale by two of three independent raters (RH, FO’M, or KS). Interrater reliability (ICC) was 0.71 for scenario 1 and 0.86 for scenario 2. We computed the sum of the total weighted scores of the two scenario questions.

Professional attitude was assessed with an 8-item questionnaire. Items were scored on a five-point scale (1 = strongly agree; 5 = strongly disagree) and the total score ranged from 8 to 40. The questions reflected the health care worker’s non-judgmental attitude towards helping other people. An example of a question is: “When a man caused a fatal accident because he was drunk, I would not offer any help to this person.” Cronbach’s alpha for the professional attitude questionnaire was *α* = 0.54 at baseline, *α* = 0.47 at post-assessment, and *α* = 0.48 at follow-up.

Confidence in taking care of people who have experienced a crisis or difficult event was assessed using a 6-item questionnaire developed for this study asking participants to rate how confident they feel in taking care of a distressed person who has experienced a crisis or difficult event, with items such as feeling confident about: “Knowing the needs of people affected by crisis.” Items were scored on a five-point scale (1 = not at all confident; 5 = extremely confident). Total scores ranged from 6 to 30 and Cronbach’s alpha for the confidence questionnaire was *α* = 0.78 at baseline, *α* = 0.73 at post-assessment, and *α* = 0.81 at follow-up.

Professional quality of life was assessed with 10 items selected from the Professional Quality of Life Scale (ProQOL-5; [16]). The ProQOL-5 assesses one’s perceived quality of life in relation to working as a helper and includes both the positive and negative aspects of this work. The ProQOL has good psychometric properties and has been administered worldwide, including in low and middle income countries. Pilot-testing of the full 30-item scale with health care workers in a district not involved in the study identified a number of difficulties in understanding and responding to 20 of the items, so these were removed. The 10 selected items were 6 items from the Compassion Fatigue scale (items 3, 12, 20, 22, 24, and 30) and 4 items from the Burnout Scale (items 2, 3, 5, and 7). Responses were given on a 5-point Likert scale where 1 = never and 5 = very often. The total score on the 10 selected items ranged from 10 to 50, where higher scores indicated a better professional quality of life. Cronbach’s alpha for the professional quality of life items was *α* = 0.61 at baseline, *α* = 0.59 at post-assessment, and *α* = 0.63 at follow-up.

All questionnaires are available from the first author upon request.

### 2.3. PFA Training

One-day face-to-face PFA group trainings were conducted for the health care workers in the Peripheral Health Units (PHUs) allocated to the “PFA group.” PFA trainings were overseen by John Snow Institute (JSI) and University of Makeni, and were provided by mental health nurses who had participated in a one-day Training of Trainers (ToT) delivered by the WHO two months earlier. PFA trainings were based on a PFA ToT manual adapted by the WHO Mental Health focal person for Sierra Leone (Dr. Florence Baingana) which included elements of mental health awareness along with PFA training based on the PFA Facilitators’ Manual for Orienting Field Workers [2]. PFA trainings are considered to enhance knowledge and competence with regard to several domains, including establishing initial contact, helping responsibly, preparation for helping, observing the situation, listening, referral, and self-care [10]. In this training, the following topics were covered: (1) explaining important terms (mental health, mental disorder, psychosocial support and psychosocial disorder); (2) understanding reactions to traumatic and stressful events; (3) understanding PFA; (4) understanding sources and signs of stress; (5) self-care; (6) providing PFA-prepare for your role, look, listen and link; (7) ending your assistance; (8) practicing PFA with role-play. 

A total of 11 trainings with an average of 19 participants were delivered between June 12 and 22, 2017, which was on average 11.1 (SD = 1.4) weeks following the baseline assessment. The participants in the control arm received the PFA training after the completion of the study, between October 31 and November 2, 2017.

A sample of six of the 11 trainings were observed by an independent research team member and rated for fidelity against 43 key features of PFA training using a structured assessment form for the fidelity of the training to the original PFA training model (Appendix A). Trainings showed acceptable-to-excellent fidelity with scores ranging from 72% to 95% adherence to key features of PFA training. 

### 2.4. Analyses

Baseline data were analysed in IBM SPSS 24 using independent t-test and chi-square analyses. For the primary and secondary endpoint analyses, linear mixed models, accounting for missing values, were employed using the lme4 package in R statistics version 3.4.3 (completers analysis; *n* = 333; see Figure 1). Seventy-one (34.5%) people in the program group did not attend the PFA training, mainly due to practical reasons including heavy rainfall in Sierra Leone during the days of the trainings. In the control group, four (2.0%) people received PFA when they should not have received it. We performed both completers and intention-to-treat analysis, but we judged the completers analysis as the main outcome analysis since we considered it most relevant to examine the training effects of PFA training in individuals who were actually trained. In addition, we considered attrition bias unlikely since the reason for most people not having received the condition they were assigned to (PFA or control) was external (extreme weather conditions). The mixed model included training (PFA vs. control), time, interaction between training and time as fixed effects, subject as random effect, and PHU as fixed effect. The mean difference between two groups at each time point, together with its 95% confidence interval (CI), was derived from the mixed model. Cohen’s *d* effect sizes were computed for the completers sample by dividing the baseline-adjusted estimated marginal mean difference between the control and PFA group by the pooled standard deviation. *p*-values of *p* < 0.05 were assumed to indicate statistical significance.

## 3. Results

### 3.1. Participants

Of the 143 PHUs approached, a total number of 129 PHUs (408 participants) were randomized into receiving PFA training (63 PHUs; 206 participants) or control (66 PHUs; 202 participants). Of the 206 participants who were allocated to PFA training, 135 (65.5%) received PFA, whereas 71 (34.5%) did not receive PFA due to factors including heavy rainfall during the days of the trainings. Chi-square tests showed that this occurred mainly in the Western Rural district, where 25 (73.5%) participants of the 31 participants assigned to PFA did not receive the PFA training. In the other districts, the numbers of participants not receiving the PFA training were five (Bombali; 15.6%), nine (Western Urban; 20.5%), five (Port Loko; 21.7%), 12 (Kenema; 31.6%), and 15 (Kailahun; 42.9%). Of the 198 participants who were allocated to control, four participants (1.9%) received the training. Participants not receiving the allocated condition (PFA or control) were excluded from the completers analysis. Therefore, the final (completers) sample consisted of 333 participants: 135 PFA and 198 control participants (Figure 1).

The original sample consisted of more participants with senior school education level (26.7% in original sample vs. 12.6% in completers sample; *χ*^2^(5) = 11.22; *p* = 0.047). There were no other differences between the original sample and the completers sample in terms of baseline characteristics or outcomes at the three assessments. 

Baseline characteristics for the PFA and control group are presented in Table 1. Independent t-tests and chi-square tests did not indicate significant differences between the two groups for any of the baseline characteristics. However, there was a trend (*p* = 0.07) indicating slightly more males in the PFA group than in the control group. In addition, there were no differences in baseline scores between the PFA and control group on knowledge of appropriate psychosocial responses (*p* = 0.91), scenario score examining understanding of PFA skills (*p* = 0.43), professional quality of life (*p* = 0.83), confidence (*p* = 0.90), and professional attitude (*p* = 0.49).

### 3.2. Main Outcomes

Linear mixed models in the sample of participants who received the allocated intervention showed that for overall knowledge of appropriate psychosocial responses we found a significant effect of time, which was moderated by condition (χ^2^(2) = 28.63; *p* < 0.0001). In the PFA group, knowledge about appropriate psychosocial responses increased relative to the control group. Post-hoc contrasts showed a medium to large effect size at the post-PFA assessment (mean estimated difference 1.73; d = 0.50; t(486.01) = 4.54; *p* < 0.001) and a medium effect size at the follow-up (mean estimated difference 1.54; d = 0.43; t(329.28) = 3.87; *p* = 0.001) (Table 2). 

In addition, PFA participants showed a larger increase in scores on the scenario score examining their understanding of application of PFA skills between baseline and follow-up (χ^2^(2) = 8.76; *p* = 0.01). Post-hoc contrasts showed that this difference was significant with a medium effect size at six-months follow-up (mean estimated difference 0.65; d = 0.38; t(372.84) = 3.80; *p* = 0.0002) but not at the post-assessment (mean estimated difference 0.19; d = 0.04; t(596.59) = 1.13; *p* = 0.26).

Similar, but less pronounced, effects were found when all originally randomized participants were included in the analyses (*n* = 408). See Appendix A for the results of these analyses.

### 3.3. Secondary Outcomes: Confidence, Professional Attitude and Professional Quality of Life

Linear mixed models did not show significant overall differences over time between PFA and control for professional attitude (*p* = 0.20), confidence (*p* = 0.64), and professional quality of life (*p* = 0.63) (Table 2). Post-hoc contrasts did show that at six-months follow-up, participants in the PFA group had higher scores on professional attitude than control participants (mean estimated difference 1.26; d = 0.23; t(354.88) = 2.11; *p* = 0.04). Largely similar, but slightly less pronounced, effects were found when all originally randomized participants were included in the analyses (*n* = 408; Appendix A). The only exception was that the post-hoc significant difference at follow-up for professional attitude between PFA and control was absent in the ITT sample (*p* = 0.17) (Appendix A).

## 4. Discussion

Our study evaluated whether PFA training, administered in community-based peripheral health care services in post-Ebola Sierra Leone, effectively impacted acquisition and retention of knowledge and understanding of appropriate psychosocial responses to individuals exposed to acute disaster and crisis, and improved knowledge on skills of how to apply these strategies. In addition, we examined whether PFA training improved professional attitude, confidence in being a helper, and professional quality of life. The results showed that a one-day PFA training following established guidelines effectively improved knowledge and understanding about psychosocial support strategies, both at three months and six months after the baseline assessment. We also found that at approximately six months, staff who received this PFA training showed greater understanding of applying psychosocial support strategies in response to scenarios of patients affected by acute crisis. That this effect was apparent only at follow-up rather than immediately post-training may suggest that the opportunity to put learning into practice was key in establishing this capability.

The benefits of PFA training were identified by the measures that directly reflected the content of the PFA training such as factual knowledge and understanding of PFA principles, of knowledge on how PFA should be applied in practice, and of roles and responsibilities of PFA helpers. The overall effects were, however, relatively modest (the PFA group had only an estimated 1.73 (at post-assessment) to 1.54 (at follow-up) points difference in scores as compared to control on a scoring scale of 1 to 26). Nonetheless, it is promising that the effects remained up to six months after the training. 

Against our expectations, we did not find an effect of PFA training on confidence related to providing psychosocial support in acute crisis situations or on professional quality of life, which included items on compassion fatigue and burn out. It is possible that these outcomes are more strongly associated with other factors in the professional life of PHU staff members, such as the type of workplace, years of job experience, and staff support [17,18]. To the extent that training may impact these outcomes, more extensive inputs are likely required, including the provision of supportive supervision [15]. It is also clear from our earlier work [15] that providing psychosocial support in line with PFA principles requires restraint in the use of common responses such as reassurance and advice-giving, and engaging in a new style of interaction may initially promote uneasiness in helpers.

The study was the first cRCT to evaluate the effects of PFA training on the ability of helpers to provide psychosocial support. An important strength as compared to previous uncontrolled studies [9,13] is that the design allowed us to conclude that the positive effects were attributable to PFA training, instead of, for instance, to expectations of the researchers or a-priori differences between the helpers receiving PFA training and those who would not. Other strengths include the large sample size, independent ratings of outcomes, and the independent checks of fidelity to the training. Limitations are the use of the same questionnaire at all three assessments. It is possible that participants may have “trained to the test” by answering the same questions at all three timepoints. This could have led to improvements over time across both groups and reduced sensitivity to detect differences between groups. Finally, we did not use behavioral outcomes to determine PFA’s training effects. Although challenging, it is clearly vital to measure the quality of psychosocial support that is actually provided by participants after PFA training, for instance by scoring the behaviors of helpers using video recordings or practice cases.

A crucial additional step in establishing the evidence for PFA is evaluating the benefits for adversity-affected individuals of receiving support from helpers trained in PFA. No such studies have been completed to date. One explanation may be that PFA is usually implemented in contexts that are challenging for conducting rigorous research, such as humanitarian crisis and disaster settings [19]. Furthermore, prevention studies are time-consuming and expensive, since they may require hundreds to thousands of respondents because of small effects sizes [20]. Another reason why these studies may not yet have been conducted is the lack of consensus among researchers and field workers about the outcomes that PFA should achieve. According to Forbes and colleagues [12] it is reasonable to expect improved psychological adjustment as well as benefits in improved functioning or reduced time off work, and to improve access to appropriate available care where needed. Others, however, stress that PFA is not aimed at preventing or reducing psychiatric symptoms, which need to be addressed by mental health care professionals [21]. However, these challenges should not be taken to comprise an unassailable barrier for the work required. Studies of case management strategies for the management of depression [22] and advocacy interventions for intimate partner violence [23] provide exemplars of the sort of multi-indicator public mental health approach required.

## 5. Conclusions

This study suggests that for organizations involved in rolling out PFA across settings worldwide, PFA trainings can strengthen capacity for psychosocial support in contexts of disaster and humanitarian crisis. This is welcome given the general lack of effective early interventions for individuals exposed to acute traumatic events and crisis [24]. Nevertheless, randomized controlled evaluation of the psychosocial effects of PFA on those affected by traacute crisis remains imperative, even though large-scale roll out of PFA across crisis situations worldwide is already a reality.

## Figures and Tables

**Figure 1 ijerph-17-00484-f001:**
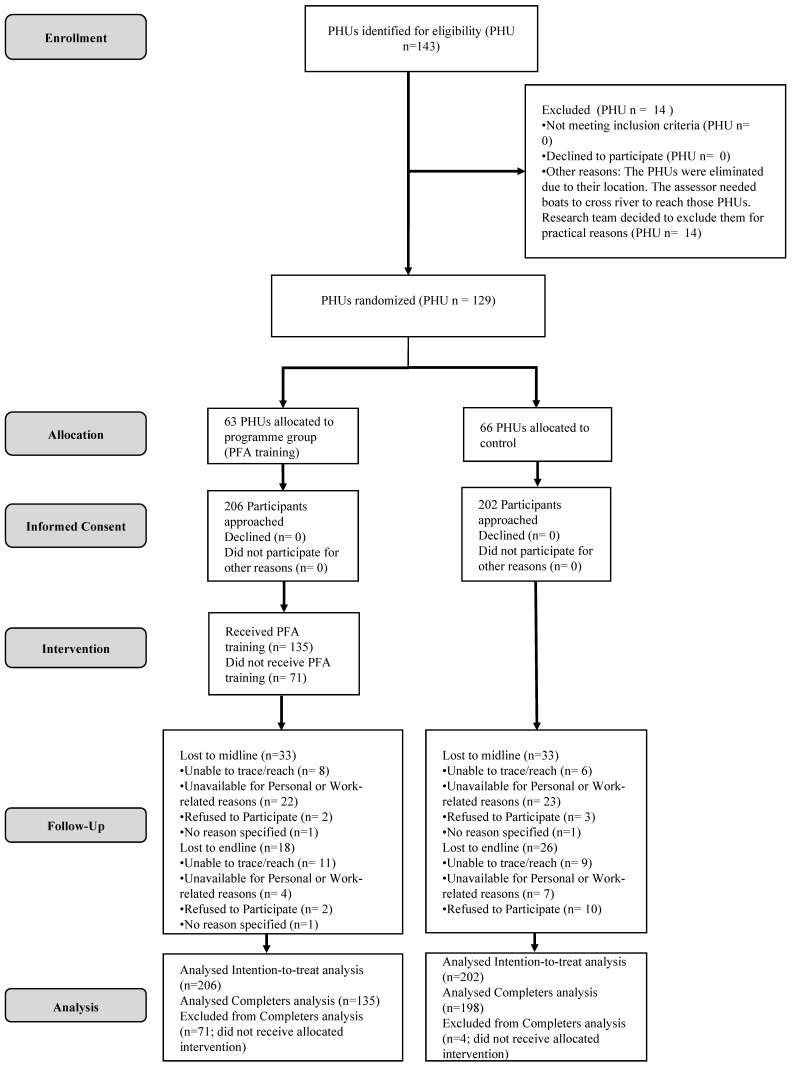
Flow chart.

**Table 1 ijerph-17-00484-t001:** Baseline characteristics (*n* = 408).

	PFA (*n* = 202)	Control (*n* = 206)			
**Characteristic**	**Mean (SD)**	**Mean (SD)**	**t**	**df**	***p***
Age (mean, SD)	39.5 (9.26)	38.5 (9.0)	1.14	404	0.25
Work experience in years (mean, SD)	7.18 (6.38)	7.88 (7.48)	−1.006	403	0.32
	***n*** **(%)**	***n*** **(%)**	***Χ*^2^**	**df**	***p***
Gender ^1^			3.30	1	0.07
Female	165 (80.5)	176 (87.1)			
Male	40 (19.5)	26 (12.9)			
Religion ^2^			0.72	1	0.40
Christian	117 (57.9)	108 (53.7)			
Muslim	85 (42.1)	93 (46.3)			
Education			1.76	5	0.88
Junior Secondary School	2 (1)	2 (1)			
Senior School	28 (13.6)	34 (16.8)			
Certificate	154 (74.8)	148 (73.3)			
Diploma	19 (9.2)	14 (6.9)			
Undergraduate/Graduate	3 (1.5)	4 (2)			
Profession			5.09	4	0.28
Nurse	73 (35.4)	89 (44.1)			
Community Health Worker	20 (9.7)	12 (5.9)			
Midwife	15 (7.3)	16 (7.9)			
Maternal health assistant	80 (38.8)	73 (36.1)			
Other (vaccinator, lab assistant, etc.)	18 (8.7)	12 (5.9)			
Direct contact with people in distress			0.34	1	0.56
Yes	200 (97.1)	194 (96.0)			
No	6 (2.9)	8 (4.0)			

PFA = Psychological First Aid. ^1^ Data from 407 participants; ^2^ Data from 403 participants.

**Table 2 ijerph-17-00484-t002:** Summary statistics and results from mixed model analysis of primary and secondary outcomes (*n* = 333).

		Descriptive Statistics Mean (SD)		Cohen’s d	Mixed Model Analysis *	
Outcomes	Time Point	PFA (*n* = 135)	Control (*n* = 198)		Difference in LS Mean (95% CI)	*p*-Value
Knowledge retention score	Baseline	12.18 (3.40)	12.17 (3.05)			
	Post-assessment	14.08 (3.53)	12.34 (3.26)	0.50	1.73 (0.98–2.47)	<.0001
	Follow-up	14.17 (3.34)	12.60 (3.39)	0.43	1.54 (0.76–2.33)	0.0001
Scenario score	Baseline	2.92 (1.39)	2.80 (1.31)			
	Post-assessment	3.36 (1.52)	3.16 (1.38)	0.04	0.19 (−0.14–0.51)	0.26
	Follow-up	3.60 (1.38)	2.98 (1.38)	0.38	0.65 (0.31–0.98)	0.0002
Professional attitude	Baseline	30.66 (5.55)	30.58 (5.28)			
	Post-assessment	31.20 (4.58)	30.58 (5.28)	0.14	0.78 (−0.36–1.92)	0.19
	Follow-up	31.57 (4.84)	30.35 (5.32)	0.23	1.26 (0.09–2.42)	0.04
Confidence	Baseline	19.42 (4.38)	19.09 (4.45)			
	Post-assessment	20.25 (4.45)	19.34 (4.03)	0.10	0.76 (−0.21–1.73)	0.13
	Follow-up	19.56 (4.22)	19.20 (4.15)	0.01	0.29 (−0.65–1.24)	0.54
Professional quality of life	Baseline	37.07 (5.73)	36.36 (5.69)			
	Post-assessment	36.87 (5.52)	36.30 (5.51)	0.12	0.07 (−1.21–1.35)	0.91
	Follow-up	36.79 (6.10)	36.58 (5.52)	0.03	0.51 (−0.81–1.83)	0.45

PFA = Psychological First Aid; * The mixed model included training, time, interaction between training and time as fixed effects, subject as random effect, and PHU as fixed effect.

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
