# Peer review of "The Effect of Psychological First Aid Training on Knowledge and Understanding about Psychosocial Support Principles: A Cluster-Randomized Controlled Trial"

_ijerph, 2020, doi:10.3390/ijerph17020484_

Round 1

Reviewer 1 Report

As this paper points out we still have a paucity of articles on Psychological First Aid (PFA).  This study has certain strengths.

It has a good sample size which allows for reasonable power in the analyses.

It has a control design and multi stage follow up which is exceedingly rare.

The findings are well-described and do not exaggerate.

The literature review is mainly reasonable.  I would add more on why PFA might not be enough and its limitations.  Also, the Hobfoll 5 principles paper should be added to bring in more theoretical and consensus information on what needs to be provided.

The sample being in a region of high need where there are few MH professionals is a definite strength.

The paper is likely to be highly cited as a building block.  We should be further along on this topic empirically but the fact is we are not so this paper is important on those grounds.

Analyses and discussion of effect size and not just significance would be helpful.  How much more do the participants know then they did before could be better explained.  A significant difference might not be meaningful difference.

Reviewer 2 Report

Dear authors,

I have read your manuscript with great interest, and applaud your efforts to contribute to build evidence on the effects of PFA. In general, I think your manuscript is well written and has a logic structure.

However, I have some recommendations and questions to consider:

General

Make sure that the aim of the paper is coherent in abstract, background and that the conclusion is respondent to the aim.

Method

Concerning the following sentence: “Demographic characteristics were assessed at the start of the interviews”; what interviews? My guess is that the interviews were part of the larger mixed methods study, but this statement need a clarification. Regarding the questionnaire to evaluate pre-training knowledge on PFA; how did you develop the questionnaire? Has it been used before or was it study specific? Does the ICC calculation and interrater reliability evaluation belongs to some kind of evaluation of the questionnaire itself, or is this a misunderstanding? Is the questionnaire available somewhere? If not published, may it be requested by readers in any way? (You might consider to make this statement in a supplementary comment also) Please provide some information on how the randomization was made? I think it can be discussed if this is a classic RCT study, or a cluster RCT.

Result & Discussion

I agree that your finding that increased awareness of PFA and potential trauma did not lead to an increased P-QoL is interesting. I assume that most of the study participants had experienced the Ebola epidemic or other potentially traumatic events themselves. Can this have had influence on this fining? According to your opinion, what was the overall attitude to psychological trauma and mental distress among the study participants? Is there a possibility that stigma or similar causes may have influenced answers on for example QoL among the study participants? Did you find any association on knowledge or professional attitude in relation to if the study participant had been professionally involved with people in distress (e.g. provided PFA) or not after the training? I would have appreciated an ethical note on the fact that not all study participants received PFA training.

I can’t find any information neither in the method section nor in the discussion on drop outs, external or internal?

Author Response

Reviewer 2

Make sure that the aim of the paper is coherent in abstract, background and that the conclusion is respondent to the aim.

Authors’ response:

Across abstract, introduction and discussion, we made the following slight textual changes in order to make sure that the aim is entirely similar:

Abstract:

We evaluated the effectiveness of a one-day PFA training on on the acquisition and retention of knowledge of appropriate psychosocial responses and skills in the acute aftermath of adversity in Peripheral Health Units (PHUs) in post-Ebola Sierra Leone. Secondary outcomes were professional quality of life, confidence in supporting a distressed person and professional attitude. (page 1).

Introduction:

“The primary aim of this study was to extend the evidence base concerning PFA, by evaluating the effectiveness of a one-day PFA training on the acquisition and retention of knowledge of appropriate psychosocial responses and skills in the acute aftermath of adversity in Peripheral Health Units (PHU) staff members across Sierra Leone. Secondarily, we investigated whether PFA training improved professional quality of life, confidence in supporting a distressed person and professional attitude.” (page 3)

Discussion:

“Our study evaluated whether PFA training, administered in community-based peripheral health care services in post-Ebola Sierra Leone, effectively improved acquisition and retention of knowledge and understanding of appropriate psychosocial responses to individuals exposed to acute disaster and crisis, and improved knowledge on skills on how to apply these strategies.” (pages 10-11).

Method

Concerning the following sentence: “Demographic characteristics were assessed at the start of the interviews”; what interviews? My guess is that the interviews were part of the larger mixed methods study, but this statement need a clarification.

Authors’ response:

We agree that this might be confusing. Therefore, we changed the word “interviews” into “assessments” (page 4).

Regarding the questionnaire to evaluate pre-training knowledge on PFA; how did you develop the questionnaire? Has it been used before or was it study specific? Does the ICC calculation and interrater reliability evaluation belongs to some kind of evaluation of the questionnaire itself, or is this a misunderstanding? Is the questionnaire available somewhere? If not published, may it be requested by readers in any way? (You might consider to make this statement in a supplementary comment also)

Authors’ response:

The questionnaire was study-specific, since there are no formal tests or exams available to evaluate knowledge relevant to PFA. We have added the word “study-specific” to page 6.

We computed the ICC to see whether the raters, who scored the open-ended items of the test, agreed in terms of their ratings. So this is not meant to evaluate the test itself, but to assess the reliability of the scoring.

We will make the questionnaire available to researchers on request (page 5). Publishing them online is counter-indicated by this likely rendering it unusable in future in any other evaluative study.

Please provide some information on how the randomization was made? I think it can be discussed if this is a classic RCT study, or a cluster RCT.

Authors’ response:

It is indeed cluster-randomized controlled trial, since we randomized on the level of PHU to avoid contamination between participants. Although mentioned in the title, to improve clarity we have added the word cluster-randomized to the abstract (page 1), and added “cluster” to the methods on page 4.

Result & Discussion

I agree that your finding that increased awareness of PFA and potential trauma did not lead to an increased P-QoL is interesting. I assume that most of the study participants had experienced the Ebola epidemic or other potentially traumatic events themselves. Can this have had influence on this fining? According to your opinion, what was the overall attitude to psychological trauma and mental distress among the study participants? Is there a possibility that stigma or similar causes may have influenced answers on for example QoL among the study participants? Did you find any association on knowledge or professional attitude in relation to if the study participant had been professionally involved with people in distress (e.g. provided PFA) or not after the training?

Authors’ response:

This is an interesting suggestion. We agree that all study participants are likely to have been indirectly or directly exposed to the Ebola crisis, since Sierra Leone has been affected by widespread transmission of Ebola between 2014-2016. We did not assess involvement in the Ebola crisis, so we are not able to relate this to the outcomes of PFA training.

We have assessed whether participants as part of their job are directly exposed to individuals who experienced trauma and adversities (see Table 1). We found that this was the case for the large majority of our sample, only 14 participants out of 408 (3.4%) reported that they had not been exposed to such individuals. Therefore, we considered that analyzing these groups separately, would not be possible, because lack of statistical power.

I would have appreciated an ethical note on the fact that not all study participants received PFA training.

Authors’ response:

In fact, all participants received the PFA training. The control participants received the training after the final assessment. To make this clear, the following sentence was added to the methods section:

“The participants in the control arm received the PFA training after the completion of the study, between October 31 and November 2, 2017.” (page 6).

I can’t find any information neither in the method section nor in the discussion on drop outs, external or internal?

Authors’ response:

This is reported in the flow diagram (Figure 1). The flow diagram was erroneously omitted from the previous submission, our apologies for the inconvenience. We have now uploaded it.

Reviewer 3 Report

Summary :

This randomized controlled study evaluated the effectiveness of a one-day PFA training in PHUs in Sierra Leone (in a post-Ebola context) on knowledge, skills, professional attitude, confidence and professional quality of life. Assessment took place at baseline and at 3 and 6 months post-baseline. It showed stronger increase in PFA knowledge and understanding, as well as better responses to the scenarios at 6 months follow-up, among the experimental group (vs. the controlled group).

Comments :

Overall, this is a very well written study, with clear objectives, an appropriate methodology and interesting results and discussion.

Specific comments:

As mentioned by authors, PFA is usually intended for use by non-specialized helpers in contact with people recently impacted by distressing events (e.g. volunteers, teachers). In the current study, the effectiveness of PFA has been tested in Peripheral Health Units (PHU) staff members across Sierra Leone, more specifically in primary health care workers. Can we consider such workers as non-specialized helpers? In other words, I doubt that the results from this study may be generalizable to target populations of the PFA training.

As opposed to previous studies, this randomized study examined the pre- and post-test effects of PFA training, using an experimental and a control group, and therefore extends the evidence base concerning PFA. However, since 34.5% people in the programme group did not attend the PFA training (and 2.0% people in the control group received PFA), the completers analysis was privileged over the intention-to-treat one. Given the important cross-over between groups (i.e. people randomized as trained that, in fact, were not trained and people randomized as untrained that, in fact, were trained), the experimental group was no longer comparable with the control group. Indeed, Table 2 (based on the final sample size of 333 respondents) suggests that scenario score, confidence, and professional quality of life were different at baseline between the trained and the untrained groups. On that point, I think that p values for the comparison of scores at baseline between the two groups should be displayed in this table and discussed in the paper. In the same vein, Table 1 showing baseline characteristics should be based on the final sample (n=333 rather than n=408) to better reflect differences between groups that could have been introduced by the cross-over.

With the exception of the Professional Quality of Life Scale, few details are provided on the psychometric properties of measures used as primary and secondary outcomes. Have they been previously administered in other settings ? If not, how were they constructed (based on which literature) ?

The timeline of this study is confusing and should be clarified, including the dates of the baseline assessment, the training sessions, the post-training assessment, and the follow-up assessment. The terms “3-month post-assessment” and “6-month follow-up assessment" should be avoided, as in fact it refers to the time that elapsed between baseline assessment (in March-April) and further assessments (in June-July and October-December, respectively), rather than the time that elapsed since training (in June).

In this regard, I think that the sentence on page 10 “ The results showed that a one-day PFA training following established guidelines effectively improved knowledge and understanding about psychosocial support strategies, both at three months and six months after the trainings took place.” is wrong.

It is not clear why the baseline assessment was in March/April while the training was only in June.

Of the final (completers) sample, which was composed of 333 participants (135 PFA and 198 control participants), it is not clear how many were reached for the immediate post-training and the follow-up assessment. This should be more explicit, and possibly represented by a flowchart.

Finally, it is stated that participants were included in the study if they had adequate oral and written command of the English or Krio language. However, measures were only administered in the English language at all three time points. How did the researchers deal with participants that had adequate oral and written comment of the Krio language, but not of the English language ?

Author Response

Reviewer 3

Specific comments:

As mentioned by authors, PFA is usually intended for use by non-specialized helpers in contact with people recently impacted by distressing events (e.g. volunteers, teachers). In the current study, the effectiveness of PFA has been tested in Peripheral Health Units (PHU) staff members across Sierra Leone, more specifically in primary health care workers. Can we consider such workers as non-specialized helpers? In other words, I doubt that the results from this study may be generalizable to target populations of the PFA training.

Authors’ response:

We consider them as non-specialized in delivering mental health and psychosocial support interventions. However, to avoid any misunderstanding, we have now deleted the word “non-specialized” and reformulated the sentence as follows:

“PFA is largely intended for use by helpers in contact with people recently impacted by distressing events. They may include staff or volunteers such as health workers not specialized in mental health and psychosocial support, community health workers and teachers [2].” (page 3).

As opposed to previous studies, this randomized study examined the pre- and post-test effects of PFA training, using an experimental and a control group, and therefore extends the evidence base concerning PFA. However, since 34.5% people in the programme group did not attend the PFA training (and 2.0% people in the control group received PFA), the completers analysis was privileged over the intention-to-treat one. Given the important cross-over between groups (i.e. people randomized as trained that, in fact, were not trained and people randomized as untrained that, in fact, were trained), the experimental group was no longer comparable with the control group. Indeed, Table 2 (based on the final sample size of 333 respondents) suggests that scenario score, confidence, and professional quality of life were different at baseline between the trained and the untrained groups. On that point, I think that p values for the comparison of scores at baseline between the two groups should be displayed in this table and discussed in the paper. In the same vein, Table 1 showing baseline characteristics should be based on the final sample (n=333 rather than n=408) to better reflect differences between groups that could have been introduced by the cross-over.

Authors’ response:

In Table 1, we presented baseline scores of participants, and also presented the p-values comparing the PFA and control groups.

We are unsure why the reviewer assumes the baseline scores as presented in table 2 for scenario, professional quality of life are different between PFA and control group? The differences are very small. We have performed independent t-tests to compare them, and indeed, they were not significantly different. We added the following sentence to the results section to describe the results of the analyses comparing baseline scores and the outcome measures (page 7):

“In addition, there were no differences in baseline scores between the PFA and control group on knowledge of appropriate psychosocial responses (p=.91), scenario score examining understanding of application of PFA skills (p=.43), professional quality of life (p=.83), confidence (p=.90) and professional attitude (p=.49).”

Please note also that by using linear mixed models that included all three time points, any baseline differences on the outcome measures are accounted for in the analyses, since we are looking at differential change over time between the two study groups.

With the exception of the Professional Quality of Life Scale, few details are provided on the psychometric properties of measures used as primary and secondary outcomes. Have they been previously administered in other settings? If not, how were they constructed (based on which literature)?

Authors’ response:

These questionnaires were specifically developed for this study. The questions were not derived from other questionnaires. Note that in response to issue 3 of Reviewer 2, we have decided to make the questionnaires available on request to other researchers.

The timeline of this study is confusing and should be clarified, including the dates of the baseline assessment, the training sessions, the post-training assessment, and the follow-up assessment. The terms “3-month post-assessment” and “6-month follow-up assessment" should be avoided, as in fact it refers to the time that elapsed between baseline assessment (in March-April) and further assessments (in June-July and October-December, respectively), rather than the time that elapsed since training (in June).

Authors’ response:

We agree that this might be confusing. However, it is not possible to refer to the time since training in this study, since that would not be applicable to the control group, who did not receive the training during the study period. Therefore, we decided to report the time since baseline, as commonly done in RCTs.

In this regard, I think that the sentence on page 10 “ The results showed that a one-day PFA training following established guidelines effectively improved knowledge and understanding about psychosocial support strategies, both at three months and six months after the trainings took place.” is wrong.

Authors’ response:

Many thanks for spotting this error. We have changed “after the trainings took place” in after the baseline assessment (page 11).

It is not clear why the baseline assessment was in March/April while the training was only in June.

Authors’ response:

The assessments were carried out by an assessor team, supervised by the researchers (RH). It took some time and logistics to arrange a wave of trainings across the country and this could not be arranged before the randomization had taken place. The trainings occurred all in one wave and were not carried out by the researchers, but by independent staff of John Snow Institute (JSI).

Of the final (completers) sample, which was composed of 333 participants (135 PFA and 198 control participants), it is not clear how many were reached for the immediate post-training and the follow-up assessment. This should be more explicit, and possibly represented by a flowchart.

Authors’ response:

We agree, the flow chart is presented in Figure 1.

Finally, it is stated that participants were included in the study if they had adequate oral and written command of the English or Krio language. However, measures were only administered in the English language at all three time points. How did the researchers deal with participants that had adequate oral and written comment of the Krio language, but not of the English language? 

Authors’ response:

The health care workers participating in the study were all professionals who have been trained in English and have taken exams in English, so were all able to complete the questionnaire in English.

We have added the following sentence to the methods section to clarify this (page 4):

“Measures were administered in the English language since the health care workers participating had all adequate command of English and were trained in English.”